# Assessment of the Phenolic Profiles, Hypoglycemic Activity, and Molecular Mechanism of Different Highland Barley (*Hordeum vulgare* L.) Varieties

**DOI:** 10.3390/ijms21041175

**Published:** 2020-02-11

**Authors:** Na Deng, Bisheng Zheng, Tong Li, Rui Hai Liu

**Affiliations:** 1Overseas Expertise Introduction Center for Discipline Innovation of Food Nutrition and Human Health (111 Center), School of Food Science and Engineering, South China University of Technology, 381 Wushan Road, Guangzhou 510641, China; daisynadeng@hotmail.com; 2Guangdong ERA Food & Life Health Research Institute, Guangzhou 510670, China; 3Department of Food Science, Cornell University, Ithaca, NY 14853, USA; tl24@cornell.edu

**Keywords:** highland barley, phenolic profiles, hypoglycemic activity, molecular mechanism

## Abstract

The phenolic profiles, hypoglycemic activity, and molecular mechanism of the effect on type 2 diabetes mellitus (T2DM) of four highland barley varieties were investigated in the present study. The fundamental phenolics in highland barley were ferulic acid, naringin, and catechin, which mainly existed in bound form. These varieties showed favorable hypoglycemic activity via inhibition of α-glucosidase and α-amylase activities, enhancement of glucose consumption, glycogen accumulation and glycogen synthase 2 (GYS2) activity, and down-regulation of glucose-6-phosphatase (G6Pase) and phosphoenolpyruvate carboxykinase (PEPCK) activities. Specifically, ZQ320 variety exhibited the strongest hypoglycemic activity compared to the other varieties. Highland barley phenolics could inhibit gluconeogenesis and motivate glycogen synthesis via down-regulating the gene expression of G6Pase, PEPCK, and glycogen synthase kinase 3β (GSK3β), while activating the expression of insulin receptor substrate-1 (IRS-1), phosphatidylinositol 3 kinase (PI3K), serine/threonine kinase (Akt), GYS2, and glucose transporter type 4 (GLUT4). Therefore, phenolics from highland barley could be served as suitable candidates for therapeutic agent in T2DM to improve human health.

## 1. Introduction

Diabetes mellitus (DM), a widespread chronic metabolic disorder with high morbidity, mortality, and healthcare expenditure, is usually characterized by persistent hyperglycemia and disorders of glucose, lipid, and protein metabolism [1,2]. Pursuant to the latest statistics from the International Diabetes Federation, there are 463 million people with DM in the world in 2019, and this number is expected to be 700 million in 2045. Clinically, DM is usually classified into four main categories: type 1 diabetes mellitus, type 2 diabetes mellitus (T2DM), gestational diabetes mellitus, and others, such as type 3 diabetes mellitus [3]. T2DM is a non-insulin-dependent diabetes caused by insulin resistance (IR) and pancreatic *β*-cell dysfunction in hepatic, adipose, and muscle tissues, which accounts for more than 90% of diabetic patients [4]. Currently, the main drug therapies for T2DM patients are insulin and synthetic medications, such as *α*-glucosidase inhibitor, biguanides, thiazolidinediones, and rosiglitazone. Nonetheless, these drugs often have many undesirable adverse effects, including weight gain, drug resistance, and hypoglycemia [5]. Therefore, more investigations on novel medicines and therapies to improve treatment efficacy, reducing healthcare expenditure, and complications of T2DM are urgently warranted. Recently, the use of natural polyphenols (e.g., ginsenoside Rk3, mangiferin, and mulberry flavonoids) with low toxicity and side effects as alternative agents for the prevention and management of T2DM has aroused much attention [6,7,8].

Barley (*Hordeum vulgare* L.), a cereal crop belonging to the Poaceae family, is the fourth most consumed grain worldwide, only behind wheat, rice, and maize [9,10]. Generally, barley can be classified as hulled barley with husks, and hulless barley without husks [10]. Highland barley is a hulless barley cultivar and used as main staple food, which is widely present in highland areas worldwide, such as Qinghai-Tibet Plateau in China, Germany, and Nepal [11]. Previous researchers have shown that highland barley is rich in natural phytochemicals such as *β*-glucan (54–76 mg/L), phenolics (131–178 mg/L), flavonoids (145.5–247.4 mg catechin equivalents (CE)/100 g dry weight (DW)), and anthocyanins (0.06–8.98 mg C-3-G E/100 g DW) [11,12,13]. The additive and synergistic combinations of these phytochemicals endow highland barley with favorable health-promoting benefits such as anticancer, antilipidemic, and antidiabetic activities [9,13,14,15]. Concretely, emerging evidence supports the notion that the antidiabetic activity of highland barley was due to its rich *β*-glucan and phenolics [15,16].

Phenolics in highland barley are present in both free and bound forms. Bound phenolics are able to survive human gastrointestinal digestion and reach the colon tract, where they exert beneficial bioactivities upon released by microbial fermentation [14]. Phenolics act as scavengers and inhibitors of reactive oxygen species by donating hydrogen or electrons to reduce the damage to large biomolecules such as proteins, lipids, and DNA within cells [17]. Amongst the phenolics, ferulic acid, caffeic acid, chlorogenic acid, (+)-catechin, *p*-coumaric acid, and protocatechuic acid were detected in highland barley [9,12,16]. These identified phenolics showed potential antioxidant, anticancer, and hypoglycemic activities according to previous studies [12,18,19,20]. Previous findings have shown that ferulic acid could maintain glucose homeostasis in T2DM models via modulating the insulin receptor substrate-1 (IRS-1)/phosphatidylinositol 3 kinase (PI3K)/serine/threonine kinase (Akt) signaling pathway and GLUT4 translocation [21,22]. Pu et al. [23] confirmed that naringin was effective in ameliorating IR by activating IRS1/GSK3*β* pathway while blocking AMPK pathway. Daisy et al. [24] found that the glucose uptake and expression level of GLUT4 in streptozotocin-induced diabetic rats were enhanced after catechin treatment. Due to the various genetic variations, cultivars, and growth conditions of highland barley all over the world, their phenolic concentrations, bioactivities, and further utilizations will vary to some extent [10,11]. Recently, Zhu, Li, Fu, Abbasi, Zheng, and Liu [12] compared the phenolic contents, antioxidant, and antiproliferative activities of four highland barley varieties; Ramakrishna, Sarkar, Schwarz, and Shetty [16] evaluated the chemical-based hypoglycemic activities of phenolics extracted from 13 barley cultivars. They both found that there were significant differences between different cultivars.

Nowadays, some investigations have focused on phytochemicals and bioactivities of highland barley, but little is known about the hypoglycemic effect and molecular mechanism of phenolics from different highland barley varieties, especially the cellular hypoglycemic capacity. Therefore, the present study aimed to evaluate the phenolic profiles and hypoglycemic activity of four highland barley varieties both in free and bound fractions, and explore the underlying molecular mechanism on the treatment of T2DM.

## 2. Results and Discussion

### 2.1. Moisture Content of Highland Barley Varieties

The moisture content of ZQ320, ZQ2000, BQ, and HQK was 12.66% ± 0.05%, 11.95% ± 0.05%, 14.44% ± 0.03%, and 12.79% ± 0.10%, respectively (Table 1). The coefficient of variation was very low. It can therefore be speculated that there was little interference to the data below due to the insignificant fluctuations of moisture content amongst highland barley varieties. Therefore, results in this paper were expressed as DW to represent data more authentically.

### 2.2. Phenolic, Flavonoid, and Anthocyanin Contents of Highland Barley Varieties

The free, bound, and total phenolic contents of highland barley varieties were depicted in Figure 1A. On the whole, ZQ320 had the highest total phenolic content (415.0 ± 39.6 mg gallic acid equivalents (GAE)/100 g DW), followed by HQK (392.7 ± 39.8 mg GAE/100 g DW), ZQ2000 (388.4 ± 3.4 mg GAE/100 g DW), and BQ (384.6 ± 41.5 mg GAE/100 g DW), respectively. The free and bound phenolic contents ranged from 123.3 ± 9.1 to 182.0 ± 24.4 mg GAE/100 g DW, and from 210.7 ± 16.4 to 265.2 ± 21.0 mg GAE/100 g DW, respectively. Additionally, the bound fraction accounted for 53.76–68.26% of total phenolics amongst all varieties, which inferred the bound phenolics as the dominative fraction corresponding to the free one. The free phenolic content was highest in the black variety (HQK), while the bound and total phenolics were both highest in the white variety (ZQ320). The total phenolic content was consistent with the previous study (333.9–460.8 GAE/100 g DW), while the percentage contribution of bound fraction to total phenolics was slightly higher than that research (38.8–49.7%) [12]. Jood and Kalra [10] reported that the polyphenols of 12 hulless and hulled barley cultivars were between 322 ± 5.28 and 625 ± 4.09 mg/100 g DW. These discrepancies might stem from differences in species, varieties, climatic conditions, and storage. Furthermore, the phenolic content significantly exceeded other grains, such as brown rice (234.7–276.0 mg GAE/100 g DW) [25] and adlay (59.30–76.04 mg GAE/100 g DW) [26].

Flavonoids, belonging to the phenolics in whole grains, exhibit favorable antioxidant and anticancer activities [27]. As shown in Figure 1B, the descending order of total flavonoid content was: ZQ320 (181.1 ± 12.0 mg CE/100 g DW), ZQ2000 (170.9 ± 11.6 mg CE/100 g DW), HQK (144.6 ± 14.0 mg CE/100 g DW), and BQ (108.8 ± 10.4 mg CE/100 g DW). The free and bound flavonoid contents varied from 29.25 ± 3.39 to 42.36 ± 3.99 mg CE/100 g DW, and from 79.58 ± 7.16 to 142.6 ± 11.8 mg CE/100 g DW, respectively. Obviously, white varieties (ZQ320 and ZQ2000) had higher bound and total flavonoids than black (HQK) and blue varieties (BQ) (*p* < 0.05), while the black variety had the highest free flavonoids. Moreover, bound fraction contributed at least 70.67% to total flavonoids amongst the four varieties. This phenomenon indicated the predominant contribution of bound fraction to total flavonoids, which showed a similar trend with phenolics. However, Zhu, Li, Fu, Abbasi, Zheng, and Liu [12] discovered that the total flavonoid content and the percentage contribution of bound fraction to total flavonoids in four highland barley were 145.5–247.4 mg CE/100 g DW and 26.9–44.0%, respectively. Shen, Zhang, Cheng, Wang, Qian, and Qi [9] found that the total flavonoid content of black highland barley was 16.22 mg RT/g. Diverse varieties, climatic conditions, storage, and methods of extraction and determination may be the main reasons for these differences. Additionally, the total flavonoid content was slightly lower than brown rice (154.1–235.6 mg CE/100 g DW) [25], but obviously higher than corn, wheat, and oat (1.16–1.68 μmol CE/g grain) [28].

Based on a micromole basis, contributions of total flavonoids to total phenolics were ranked as follows (Figure 1C): ZQ2000 (25.77% ± 1.53%) > ZQ320 (25.63% ± 0.78%) > HQK (21.64% ± 1.96%) > BQ (16.60% ± 0.23%). The contributions of free flavonoids to free phenolics and bound flavonoids to bound phenolics were 10.51–16.56% and 21.23–30.36%, respectively. The highest contributions of flavonoids to phenolics of free, bound, and total fractions were found in ZQ2000, ZQ320, and ZQ2000, respectively (*p* < 0.05), both belonging to the white variety. These data inferred flavonoids as the dominant phenolics in highland barley, which was in accordance with reported research [12,27]. Closely agreed to flavonoid and phenolic contents aforementioned, bound fractions also had higher contribution compared to that of free forms. Bound phenolics in grains could survive gastrointestinal digestion, then released by microbial fermentation in the colon to impart potential health-promoting benefits [14]. Therefore, the higher ratio of bound phenolics in highland barley might have similar application and indicate its further utilization.

Anthocyanin is a subclass of flavonoids, which provides various colors to fruits, vegetables and grains. Overall, anthocyanin was detected only in the free extracts whose contents ranged between 0.42 ± 0.08 and 9.24 ± 0.79 mg C-3-G E/100 g DW (Table 1). The highest concentration of anthocyanin was found in HQK (*p* < 0.05) in contrast with the other varieties; this might be by virtue of the black color of the HQK variety. Consequently, the black and blue varieties contain more anthocyanins compared to the other two white varieties. These results were consistent with the report about the total anthocyanin content (0.06–8.98 mg C-3-G E/100 g DW) in different highland barley varieties [12]. However, pervious research found that the total anthocyanin content of black highland barley was 3.76 mg C-3-O-G/g [9], this diversity may be chiefly attributed to the varieties and climatic conditions. To our knowledge, phenolic compounds are the products of secondary metabolism in plants, which are one of the major factors for plants to resist pathogens, parasites, and predators [14]. In the present study, the average phenolic and flavonoid contents of highland barley were 395.2 ± 13.6 mg GAE/100 g DW and 151.3 ± 32.3 mg CE/100 g DW, respectively.

### 2.3. Phenolic Profiles of Highland Barley Varieties

Overall, seven phenolics comprising four phenolic acids and three flavonoids were detected in highland barley using HPLC-PAD (Appendix A and Table 2). The limit of detection (LOD) and limit of quantitation (LOQ) of the method were 0.012–0.076 mg/mL and 0.037–0.230 mg/mL, respectively, and the spiked recoveries of standards were between 82.86 ± 3.21 to 102.41 ± 1.37% (Appendix A). The phenolic compositions of the four highland barley varieties were similar. Ferulic acid, naringin, and catechin were the dominant phenolics, which mainly existed in bound form. The contents of naringin and catechin showed significant differences (*p* < 0.05) amongst the four varieties, while that of ferulic acid had no obvious difference (*p* > 0.05). In concrete, total ferulic acid content was in a decreasing order of ZQ2000 (54.65 ± 0.77 mg/100 g DW) > BQ (51.18 ± 1.53 mg/100 g DW) > HQK (51.13 ± 2.76 mg/100 g DW) > ZQ320 (50.16 ± 1.45 mg/100 g DW). Naringin and catechin varied between 7.32 ± 0.32 (HQK) and 9.84 ± 0.23 (BQ), and between 2.47 ± 0.01 (ZQ2000) and 4.72 ± 0.01 (BQ), respectively. Amongst these phenolics, gallic acid (0.13–1.38 mg/100 g DW) was only discovered in the free extracts, quercetin (1.40–3.00 mg/100 g DW) was only observed in the bound extracts. In addition, the total contents of protocatechuic acid and chlorogenic acid ranged from 1.53 ± 0.04 (ZQ2000) to 4.17 ± 0.06 (HQK), and from 0.97 ± 0.03 (BQ) to 1.96 ± 0.02 (ZQ320), respectively. Slightly out of line with the above results, Zhu, Li, Fu, Abbasi, Zheng and Liu [12] found that the contents of total ferulic acid, catechin and chlorogenic acid were 48.14–67.53, 12.16–20.53, and 23.92–48.55 mg/100 g DW, respectively. This may be due to variation in their genetics, environmental factors, and methodological differences. These data imply that ferulic acid was the fundamental phenolics in highland barley and mostly presented in the bound fraction, which was in great concordance with former article [28]. Taken together, these chemical extraction results partly suggested highland barley as an abundant source of phenolics. With high levels of phenolic and flavonoid in highland barley, strongly indicating its potential biological activities for human wellness. Therefore, the hypoglycemic activity of four highland barley varieties both in chemical and cellular assays and the molecular mechanism were investigated below.

### 2.4. Chemical-based Hypoglycemic Activity of Highland Barley Varieties

#### 2.4.1. α-Glucosidase Inhibitory Effects

Evidently, *α*-amylase, located in the pancreas and salivary glands, can degrade dietary sugars such as starch into oligosaccharides. Subsequently, *α*-glucosidase secreted from small intestinal hydrolyzes oligosaccharides into monosaccharides which are absorbed into the blood circulation to increase the blood glucose levels [29]. Therefore, the postprandial blood sugar content in T2DM could be favorably controlled by suppressing the *α*-glucosidase and *α*-amylase activities, thus managing and treating T2DM [30]. Acarbose, a competitive inhibitor of *α*-glucosidase and *α*-amylase, was used as a positive control in present study.

The inhibitions of *α*-glucosidase and *α*-amylase amongst four highland barley varieties were given in Figure 2 and Table 3. Overall, the *α*-glucosidase activity was significantly suppressed by acarbose, four highland barley varieties and three phenolic compounds in a concentration-dependent manner. At the concentration of 100 mg/mL, the inhibitory rates of ZQ320 and ZQ2000 bound extracts were 97.06% ± 1.35% and 96.29% ± 1.34%, respectively, which were higher than that of acarbose (93.44% ± 0.73%). Particularly, ZQ320 bound extract showed the highest suppressive effect on *α*-glucosidase at each concentration point. The inhibitory efficiency of inhibitor was assessed by median inhibition dose (IC_50_), and thus a lower IC_50_ indicated a stronger inhibition effect on *α*-glucosidase. The IC_50_ of acarbose was 4.62 ± 0.68 mg/mL (Table 3). According to the IC_50_ values of samples, it can be confirmed that HQK free fraction had the strongest inhibition effect, followed by BQ, and ZQ2000, and that of ZQ320 was the weakest. In terms of bound fraction, the suppression capacities were ranked as follows: ZQ320 > ZQ2000 > BQ > HQK. Notably, the bound extracts exhibited stronger inhibition effect than the free extracts among the four varieties, which suggested the bound phenolics as favorable *α*-glucosidase inhibitors. Furthermore, the IC_50_ of ferulic acid, naringin, and catechin (10.80 ± 0.90, 17.06 ± 0.62, and 8.04 ± 0.50 mg/mL, respectively) indicated that the inhibitory effect of highland barley was not caused by an individual compound but by the synergistic and additive effects of phenolics. Previous researchers suggested that ferulic acid, catechin, and quercetin could be used as competitive suppressors of *α*-glucosidase [30,31]. Tadera et al. [32] proposed that the *α*-glucosidase could be inhibited by hydroxyl substitution on the B ring or the presence of 3-OH, 4-CO and 2,3-double bond. Therefore, higher ferulic acid and quercetin contents of bound extracts than free fractions may be responsible for the stronger prohibition of *α*-glucosidase (Table 2).

#### 2.4.2. α-Amylase Inhibitory Effects

In Figure 2B, acarbose, four highland barley varieties and three phenolic compounds all possessed a dose-dependent inhibitory effect on the *α*-amylase. The suppression effect was detected only in the bound extracts. Specifically, the inhibitory rates of ZQ320 and ZQ2000 bound extracts at 50 mg/mL were 97.95% ± 1.02% and 96.74% ± 0.65%, respectively, which exceeded that of acarbose (92.73% ± 0.81%). ZQ320 bound phenolics exhibited the strongest inhibitory effect on *α*-amylase at every concentration point. Generally, a lower IC_50_ inferred a higher inhibition effect. The IC_50_ (Table 3) of samples suggested that the suppression activities of highland barley varieties were ranked as a decreasing order of ZQ320 > ZQ2000 > BQ > HQK. The IC_50_ of ZQ320 bound phenolics (9.48 ± 1.04 mg/mL) was slightly lower than that of acarbose (3.36 ± 0.47 mg/mL). Additionally, the IC_50_ of ferulic acid, naringin and catechin (7.61 ± 0.83, 49.75 ± 0.47, and 7.04 ± 0.39 mg/mL, respectively) suggested that the inhibitory effects of highland barley may be by virtue of the synergistic and additive effects of their phenolics rather than their individual phenolic compounds. These findings revealed the potential of ZQ320 bound extract as a promising *α*-amylase suppressor, overcoming the adverse effects of acarbose such as diarrhea, abdominal distension, and pain. Consequently, the inhibition effect of *α*-amylase in bound extracts showed similar trend to *α*-glucosidase, this may be chiefly related to their higher phenolic and flavonoid contents, particularly ferulic acid, naringin, and catechin. These phenomenon suggested that ZQ320 could alleviate the postprandial hyperglycemia via inhibiting the activities of *α*-glucosidase and *α*-amylase [33]. Similar findings were reported that flavonoids such as catechin could exert stronger *α*-amylase inhibitory capacity due to the hydroxylation or the existence of 2,3-double bond and C-4 ketonic group instead of other phenolics [34]. Rasouli, Hosseini-Ghazvini, Adibi, and Khodarahmi [30] found that ferulic acid and catechin could directly interact with enzyme active sites of *α*-glucosidase (Arg^407^, Asp^326^, Arg^197^ residues, etc.) and *α*-amylase (Asp^197^, Glu^233^, Asp^300^ residues, etc.) via van der Waals and hydrogen bond to suppress their enzyme activity; in addition, they concluded that the existence of OH groups in position 3 (ring C), 7 (ring A), 4′ and 5′ (ring B) was chiefly responsible for the *α*-glucosidase and *α*-amylase inhibitory effects of polyphenols. Moreover, highland barley has confirmed to exert excellent antioxidant capacity [9,12]. It can therefore be speculated that the antioxidant activity of highland barley should be one of the reasons for its *α*-glucosidase and *α*-amylase inhibitory effects. Taken together, this study demonstrated that the ZQ320 bound phenolics effectively inhibited the activities of *α*-glucosidase and *α*-amylase, which could play a promising role for the treatment of the postprandial hyperglycemia.

### 2.5. Cell-based Hypoglycemic Activity of Highland Barley Varieties

#### 2.5.1. Glucose Consumption Activity

IR is a state of hepatic, adipose, and muscle cells/tissues with declined responsiveness to consumption and utilization of glucose, which could ultimately trigger severe metabolic syndromes such as T2DM [35]. The liver is the first tissue to become IR, thus resulting in impaired blood glucose homeostasis such as promotion of glucose production and failure to achieve glycogen synthesis [36]. Therefore, the IR-HepG2 model was constructed to assess the glucose utilization and glycogen synthesis in the liver after being administrated four highland barley varieties. Metformin (Met), as a biguanide hypoglycemic drug for T2DM, was used as a positive control. As illustrated in Figure 3A, there were no apparent influences on the viability of HepG2 cells at each concentration point. Furthermore, the median cytotoxic concentration (CC_50_) of free and bound extracts were both over 300 mg/mL (Table 1), which implied that the tested sample concentrations (10–150 mg/mL) had no cytotoxicity on HepG2 cells. In Figure 3B,C, the glucose consumption in the IR-HepG2 model group (1.87 ± 0.12 mmol/L) was much lower than normal group (3.57 ± 0.32 mmol/L, *p* < 0.05), demonstrating that the IR model was induced successfully. After supplementation with highland barley free and bound extracts, the glucose utilizations were dose-dependently ascended (*p* < 0.01). Then they reached to the highest at 150 mg/mL, whose glucose utilizations were all higher than positive control group (8.04 ± 0.31 mmol/L). Compared to the model group, the increase of glucose consumption in free extracts at 150 mg/mL were ranked as a decreasing order of ZQ320 (4.43-fold) > HQK (3.80-fold) > BQ (3.76-fold) > ZQ2000 (3.48-fold); while the bound fractions were ZQ320 (3.90-fold) > ZQ2000 (3.85-fold) > HQK (3.81-fold) > BQ (3.60-fold). There were obvious differences between free and bound fractions in ZQ320 and ZQ2000 varieties (*p* < 0.05), but were inverse to that of the BQ and HQK varieties (*p* > 0.05). Evidently, both free and bound phenolics of ZQ320 variety exhibited the strongest glucose consumption capacity in contrast with the other varieties (*p* < 0.05), indicating its hypoglycemic potential for the treatment of T2DM.

#### 2.5.2. Glycogen Accumulation Capacity

According to the data of glucose consumption, 50, 100, and 150 mg/mL of highland barley extracts were chosen for the below experiments. In order to further confirm the enhancement of glucose utilization, the glycogen content was determined (Figure 3D,E). There was a significant diminution of glycogen content in model group compared to the normal group (*p* < 0.05). After the intervention of highland barley extracts, the glycogen contents were concentration-dependently ascended and reached to the maximum at 150 mg/mL extracts. In concrete, the increments of glycogen level in free extracts at 150 mg/mL were listed as ZQ320 (1.17 times) > HQK (1.01 times) > BQ (0.84 times) > ZQ2000 (0.81 times) compared with the model group; while that of bound forms were ZQ320 (1.66-fold) > HQK (1.38-fold) > ZQ2000 (1.20-fold) > BQ (1.09-fold). There were notable differences between free and bound fractions among four highland barley varieties (*p* < 0.01). Consequently, both free and bound phenolics of ZQ320 variety showed the highest glycogen synthesis ability compared with the other varieties (*p* < 0.05), which was in great compliance with that of glucose consumption (Figure 3B,C). Therefore, the ZQ320 variety was selected for the following research to dissect its molecular mechanism. These data suggested that highland barley phenolics could promote glucose consumption and then transform it into glycogen within cells. Interestingly, in contrast with model group, the increase of glycogen content (1.00-fold) in the Met-treated group was not proportional to that of glucose consumption (3.30-fold). Met accelerates glucose uptake through inhibiting hepatic gluconeogenesis instead of accelerating glycogen synthesis may interpret this phenomenon. Consistent with our research, Vinayagam et al. [37] reviewed that the mechanisms of ferulic acid on the control of DM were ascribed to the decrease of blood glucose, improvement of insulin sensitivity, and modulation of oxidative stress. Altogether, these data suggested the beneficial effect of highland barley phenolics on enhancing peripheral glucose utilization by promoting glucose consumption and glycogen synthesis in IR model.

#### 2.5.3. Glucose-6-phosphatase (G6Pase), Phosphoenolpyruvate Carboxykinase (PEPCK), and Glycogen Synthase 2 (GYS2) Activities

G6Pase, PEPCK, and GYS2 play crucial roles in glucose homeostasis because they were major rate-limiting enzymes of gluconeogenesis (G6Pase and PEPCK) and glycogen synthesis (GYS2) [38]. Deficiency of these key enzymes in DM triggers the failure to glucose homeostasis. Therefore, ZQ320 free and bound phenolics were chosen to further explore G6Pase, PEPCK, and GYS2 activities in liver via IR-HepG2 model (Figure 4A,C). In contrast with the normal group, the IR model group significantly increased activities of G6Pase and PEPCK, and diminished GYS2 activity in HepG2 cells (*p* < 0.05). When supplemented with Met (positive control), the G6Pase and PEPCK activities of cells were observably decreased by 0.40 and 0.55-fold, respectively, and GYS2 activity were increased by 0.38-fold (*p* < 0.05) compared with those of model group. After the intervention of free and bound highland barley phenolics, the activities of G6Pase and PEPCK were reduced dose-dependently, whereas GYS2 activity were obviously ascended in a concentration-dependent pattern. At 150 mg/mL of bound extract, the decrease of G6Pase (0.49 times) and PEPCK activities (0.64 times) and the enhancement of GYS2 activity (0.61-fold) exceeded those of Met-treated group (*p* < 0.05), respectively. Interestingly, the GYS2 activity in Met-treated cells was not equivalent to its G6Pase and PEPCK activities because of the mechanism of action of Met, which was in line with the glycogen accumulation and glucose consumption assays described above. Similarly, previous researchers also confirmed that ferulic acid, naringin, and catechin could favorably reduce blood glucose via promoting glycogen synthesis and glucokinase activity in the liver [21,24,39].

### 2.6. Effects of Highland Barley on the Gene Expression Levels

Evidently, the IRS-1/PI3K/Akt signaling pathway is a major pathway involved in the regulation of glucose and insulin metabolism; it is also highly associated with IR in T2DM [40]. GLUT4 is a member of glucose transporter that accelerates insulin-stimulated glucose transport and inhibits gluconeogenesis [24]. In order to further explore the molecular mechanism of highland barley phenolics on hypoglycemic activity, the gene expression levels in the cells were measured by RT-qPCR (Figure 5). Overall, the expression levels of IRS-1, PI3K, Akt, GYS2, and GLUT4 in the IR-HepG2 cells were notably descended compared with the normal group (*p* < 0.05), while those of G6Pase, PEPCK, and GSK3*β* were significantly augmented. However, the administration of ZQ320 caused a remarkable restoration of these gene expressions in contrast with the model group, especially in ZQ320 bound phenolics treatment group. These results were well coincided with the expressions of these markers in Met-treated group. Taken together, the transitions of gene expressions might be due to the synergetic effects of phenolics such as ferulic acid, naringin, and catechin on IRS-1/PI3K/Akt pathway as well as GLUT4 in T2DM models [22,23,24]. Similar findings were observed for other natural plants with rich phenolics that could modulate the IRS-1/PI3K/Akt signaling pathway and mitigate IR in T2DM [41,42]. From these findings, it can be concluded that highland barley could maintain glucose homeostasis through the down-regulation of key enzymes of gluconeogenesis (G6Pase and PEPCK) and up-regulation of glycogen synthesis (GYS2) in IR-HepG2 cells. Highland barley exhibited potential hypoglycemic function by enhancing glucose consumption and glycogen accumulation because of the IRS-1/PI3K/Akt signaling pathway and GLUT4 translocation.

However, the present study did not consider intestinal absorption/metabolism, which would significantly change the proportions and types of phenolic compounds HepG2 cells would be exposed to. Therefore, the known bioavailability of the free and bound phenolics identified herein were discussed below. Achour et al. [43] observed that ferulic acid only hydrolyzed into glucuronidates after intestinal and hepatic metabolisms, and the metabolites percentage in HepG2 cells was 3.2%. Sun et al. [44] suggested that the contents of naringin and protocatechuic acid decreased, while ferulic acid increased significantly in all citrus fruits (*p* < 0.05) after in vitro gastrointestinal digestion, and the HepG2 cell uptake contents of naringin and protocatechuic acid in digested citrus samples were 448–1684 and 127–1040 μg/100 g DW, respectively. Tagliazucchi et al. [45] found that catechin would not be degraded by gastric and pancreatic digestion, while gallic acid content decreased 43.3% after pancreatic digestion. Previous reports showed that approximately 29% and 23% of quercetin, respectively, in Caco-2 and HepG2 cells were the original aglycone forms and the others were the methylated and gulucuronide/sulfate conjugates [46,47]. The previous studies confirmed that chlorogenic acid could be swiftly absorbed without structural transitions by rats and humans [48,49]. These findings confirmed that the identified free and bound phenolics in the present study could be easily absorbed by cells/bodies with slight or without structural transformations after gastrointestinal metabolisms. Therefore, the present explorations on the hypoglycemic capacity of highland barley in HepG2 cells would provide a guideline on the comprehensive utilization of phenolics in this whole grain. Furthermore, the objective of our future research is to unravel the underlying mechanism of action of highland barley phenolics in vivo on T2DM.

## 3. Materials and Methods

### 3.1. Materials and Reagents

Highland barley Zangqing 320 (ZQ320), Zangqing 2000 (ZQ2000), Dulihuang (BQ), and Heiqingke (HQK) were donated by Professor Yu Liu, in Tibet Tianyu Agricultural Technology Co., Ltd. (Shigatse, China). Gallic acid, Folin–Ciocalteu reagent, catechin hydrate, *α*-glucosidase (EC 3.2.1.20) from *Saccharomyces cerevisiae*, *α*-amylase (EC 3.2.1.1) from porcine pancreas, acarbose, *p*-nitrophenyl-*α*-D-glucopyranoside (pNPG), metformin (Met), and chromatographic grade of standards were purchased from Sigma Chemical Co. (St. Louis, MO, USA). HepG2 human liver cancer cells (ATCC HB-8065) were obtained from ATCC company (Manassas, VA, USA). Fetal bovine serum (FBS) and other cell culture reagents were purchased from Gibco Life Technologies Co. (Grand Island, NY, USA). Glucose, glycogen, and bicinchoninic acid (BCA) protein assay kits were obtained from Nanjing Jiancheng Institute of Bioengineering (Nanjing, Jiangsu, China). Human G6Pase, PEPCK, and GYS2 assay kits were purchased from Shanghai Changjin Biotechnology Co., Ltd. (Shanghai, China).

### 3.2. Determination of Moisture Content

The moisture content was measured according to the oven-dry method as explained by Guo et al. [50]. Briefly, 2 g of sample was kept in an oven at 102 °C until a constant weight. Results were presented as percent of dry weight (DW, mean ± SD, *n* = 3).

### 3.3. Extraction of Free and Bound Phenolics

Phenolics from the samples were extracted as reported before with slight modifications [17]. In brief, samples were extracted by chilled 80% acetone (1:2, *w/v*) and re-dissolved in Milli-Q-water to procure free extracts. Residues obtained were digested with 2 mol/L NaOH and acidified to pH 2 with concentrated HCl, then extracted five times by ethyl acetate. The ethyl acetate fractions were reconstituted using Milli-Q-water to yield bound extracts. Both extractions were performed in triplicates and stored at −40 °C for further analysis.

### 3.4. Determination of Phenolics, Flavonoids, and Anthocyanins

The phenolic content was measured by modified Folin-Ciocalteu approach [50] with gallic acid as a standard. The flavonoid content was tested using sodium borohydride/chloranil method [27] and catechin hydrate was used as a standard. The anthocyanin content was determined by previous spectrophotometric pH differential protocol [12,51]. Results were expressed as milligram gallic acid equivalents per 100 g DW (mg GAE/100 g DW), milligram catechin equivalents per 100 g DW (mg CE /100 g DW), and mg cyanidin 3-glucoside equivalents per 100 g DW (mg C-3-G E/100 g DW), respectively (mean ± SD, *n* = 3).

### 3.5. Identification of Phenolic Profiles

The phenolic compounds were analyzed by high performance liquid chromatography-photodiode array detector (HPLC-PAD, Waters Corp, Milford, MA) method with little modification [52], using a C_18_ column (250 mm × 4.6 mm, 5 μm) maintained at 35 °C and detected at 280 nm. The gradient of binary elution-phase (A: Milli-Q-water with 0.1% glacial acetic acid, B: acetonitrile with 0.1% glacial acetic acid) was 0–2 min 8–10% B, 2–27 min 10–30% B, 27–50 min 30–90% B, 50–51 min 90–100% B, 51–56 min 100% B, and 56–60 min 8% B at a flow rate of 0.8 mL/min. Phenolic profiles were qualified by comparing retention time and peak area of the corresponding pure compounds, and were expressed as milligrams per 100 g DW (mg/100 g DW, mean ± SD, *n* = 3).

### 3.6. Chemical-Based Hypoglycemic Activity Assays

The *α*-glucosidase and *α*-amylase inhibitory activities were measured as explained by Lordan et al. [53] and Oboh, Ademiluyi, Akinyemi, Henle, Saliu, and Schwarzenbolz [33] with minor modifications, which both used acarbose as positive control. For *α*-glucosidase inhibitory assay, appropriately diluted samples and *α*-glucosidase (0.1 U/mL) were kept at 37 °C for 10 min. Afterwards, pNPG (5 mM) was added and incubated at 37 °C for 20 min. Absorbance was monitored instantly at 405 nm by DU 730 Nucleic Acid/Protein Analyzer (BECKMAN, Kraemer Boulevard Brea, CA, USA). With regards to *α*-amylase inhibitory assay, diluted samples and *α*-amylase (1 U/mL) were kept at 37 °C for 10 min, before the addition of 1% starch solution and another incubation at 37 °C for 10 min. Then, dinitrosalicylic acid reagent was added and the mixture was kept in boiling water bath for 5 min to halt the reaction. The mixture was diluted by deionized water and absorbance was measured at 540 nm (DU 730 Nucleic Acid/Protein Analyzer, BECKMAN). Data were both assessed by median inhibition dose (IC_50_) and presented as mg/mL (mean ± SD, *n* = 3). The *α*-glucosidase and *α*-amylase inhibitory activities were calculated as followed: $\mathrm{Inhibition}\;\left(\%\right)\;=\;\left[1-\left(A_{sample}-A_{background}\right)/A_{control}\right]\;\times \;100\;$, where $A_{sample}\;$, $A_{background}$ and $A_{control}$ are the absorbances of sample, background, and control, respectively.

### 3.7. Cell-Based Hypoglycemic Activity Assays

#### 3.7.1. Cytotoxicity Assay

HepG2 cells were cultured in Dulbecco’s modified eagle medium (DMEM) containing 4.5 g/L glucose, 10% FBS, and 1% antibiotic antimycotic solution. The cytotoxicity of extracts were performed by modified methylene blue assay as described before [54]. Briefly, HepG2 cells were seeded at 4 × 10^4^ cells/well and incubated for 24 h. Growth medium was displaced by fresh medium containing various concentrations of extracts after incubation, then cells were counted by methylene blue assay. Results were evaluated by median cytotoxic concentration (CC_50_) expressed as mg/mL (mean ± SD, *n* = 3).

#### 3.7.2. Glucose Consumption Assay

The glucose consumption protocol was conducted as previously reported [55] using Met as positive control. Briefly, HepG2 cells were seeded at 3 × 10^4^ cells/well for 12 h, then DMEM medium supplemented with 33 mM glucose, and 1.5 μM insulin was added and incubated for 24 h to establish IR model. Subsequently, DMEM medium was displaced by fresh medium containing Met (80 mM) or diluted extracts. After administration for 24 h, the glucose concentration in the medium was determined by glucose assay kit. The glucose utilization of cells was calculated by initial glucose content minus residual glucose content and expressed as mmol/L (mean ± SD, *n* = 3).

#### 3.7.3. Glycogen Accumulation Assay

The glycogen accumulation protocol was conducted according to Xu et al. [56] and Met was used as positive control. HepG2 cells were seeded at 2 × 10^5^ cells/well in a 6-well microplate. After treatment by samples (refer to Section 3.7.2), the cells were collected and lysed by RIPA solution (containing 10 mM PMSF) for protein determination. The glycogen content within the cells was measured according to glycogen assay kit. Results were normalized to protein levels by the BCA protein assay kit and expressed as μg/mg (mean ± SD, *n* = 3).

#### 3.7.4. G6Pase, PEPCK, and GYS2 Activity Assays

The treatment, collection, and lysis of HepG2 cells were the same as Section 3.7.3. The G6Pase, PEPCK and GYS2 activities were tested by human Elisa kits according to instructions. Data were normalized to protein levels by the BCA reagent and presented as IU/mg protein (mean ± SD, *n* = 3).

### 3.8. Quantitative Real-Time Polymerase Chain Reaction (RT-qPCR) Assay

After administrated by samples (refer to Section 3.7.3), total RNA was extracted from the cells with TRIzol reagent. The total RNA was reverse-transcribed to cDNA and amplified using a PrimeScript™ RT Reagent kit (Takara Biotechnology, Dalian, China). RT-qPCR was performed on a CFX quantitative PCR system (Bio-Rad, CA, USA) using SYBR Green PCR Quant qPCR kits (TIANGEN Biotech, Beijing, China). The *β*-actin primer was used as an internal control. Sequences of primers (Appendix A) were designed from the National Center for Biotechnology Information (US National Library of Medicine) and synthesized by Shanghai Majorbio Bio-pharm Technology Co. Ltd. (Shanghai, China). The relative gene expression value was calculated using the 2^−∆∆*C*t^ method (mean ± SD, *n* = 3).

### 3.9. Statistical Analysis

Data were expressed as mean ± SD for triplicates. ANOVA and Tukey’s test analysis were performed by SPSS software 21.0 (SPSS Inc., Chicago, IL, USA). Significance was determined at *p* < 0.05. Dose-effect analysis was conducted using Calcusyn software 2.0 (Biosoft, Cambridge, U.K.).

## 4. Conclusions

Highland barley is a whole grain rich in natural phenolic compounds, and most of these phenolics partitioned in bound form. Highland barley phenolics exerted hypoglycemic effect by preventing *α*-glucosidase and *α*-amylase activity in vitro. Furthermore, these phenolic extracts could motivate glucose consumption and glycogen accumulation in IR-HepG2 model through IRS-1/PI3K/Akt signaling pathway and GLUT4 translocation, thus maintaining glucose homeostasis. In conclusion, this study suggested highland barley as an effective candidate for the prevention and treatment of T2DM. Therefore, the underlying mechanism of action in vivo will be sufficiently explored in our future research.

## Figures and Tables

**Figure 1 ijms-21-01175-f001:**
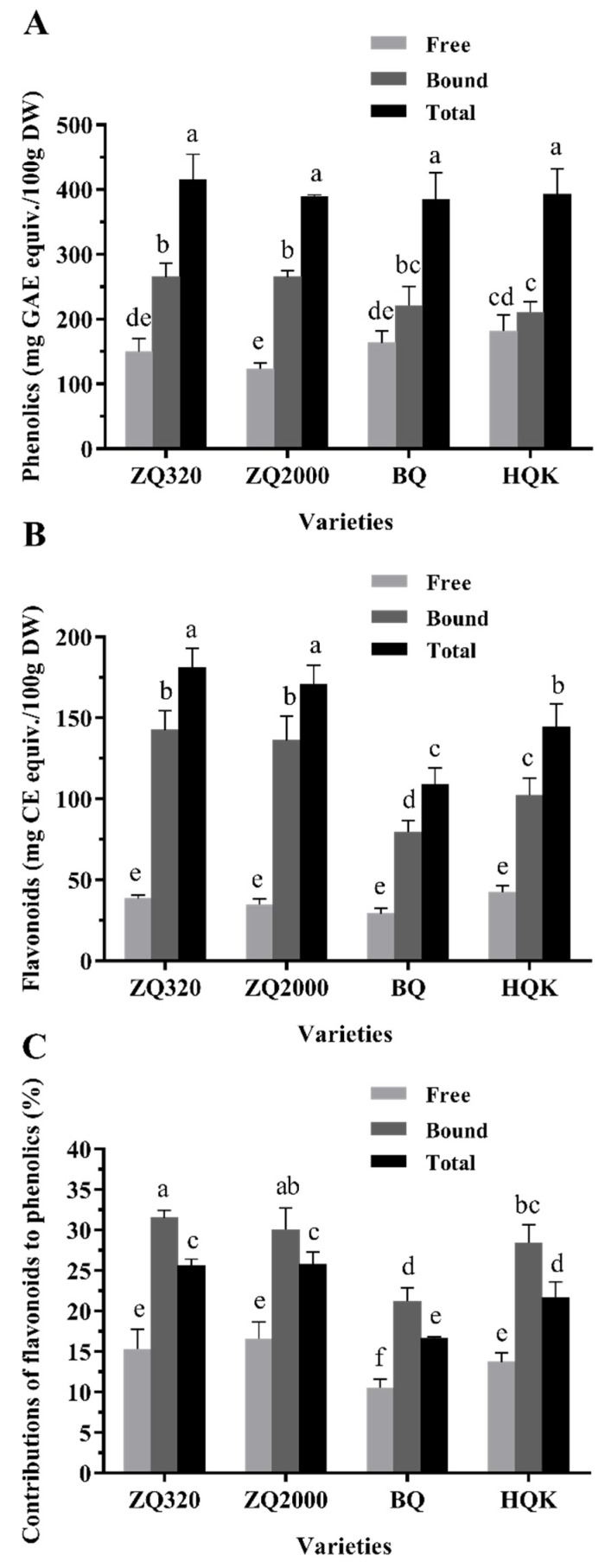
Total phenolic (**A**) and flavonoid contents (**B**) and contributions of flavonoids to phenolics (**C**) of the free, bound, and total fractions of four highland barley varieties (mean ± SD, *n* = 3). Values with different letters in each bar differ significantly at *p* < 0.05.

**Figure 2 ijms-21-01175-f002:**
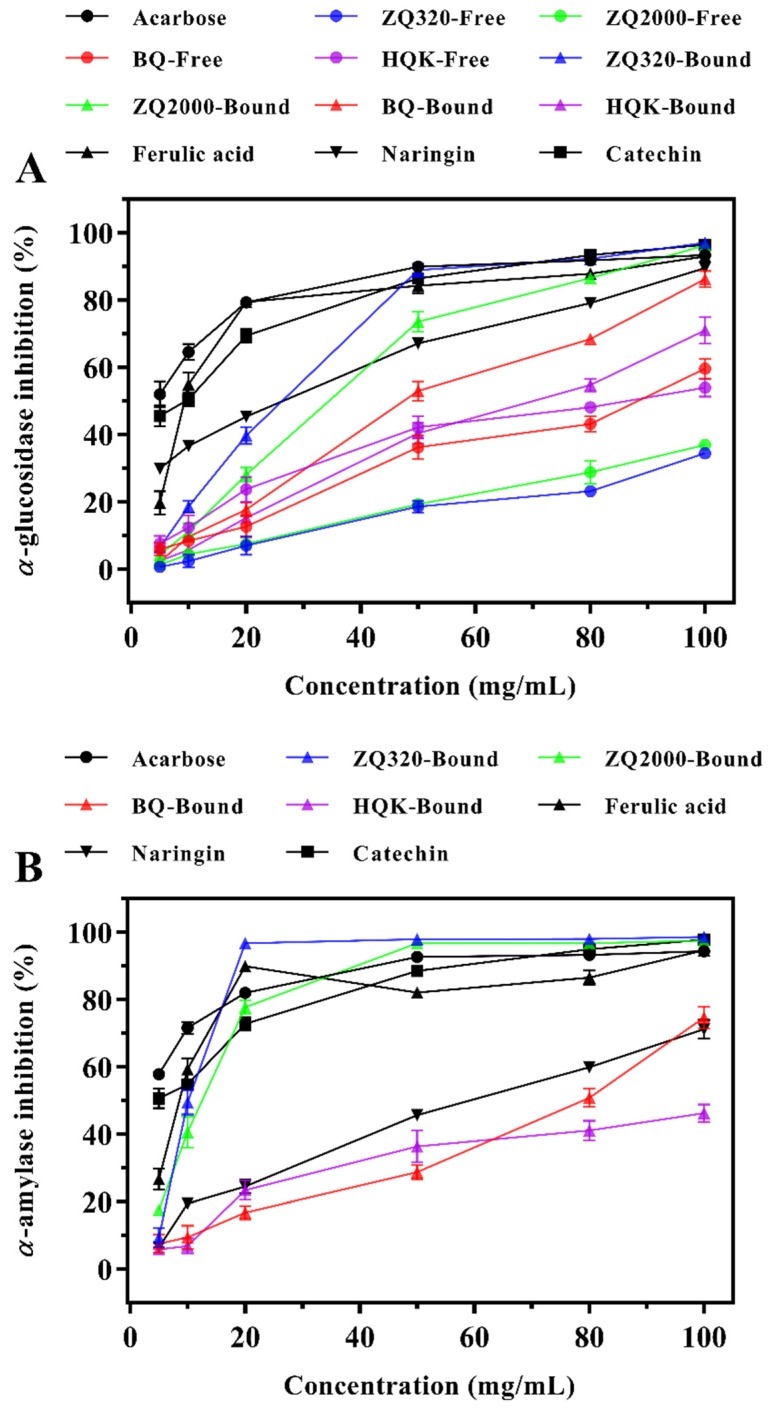
*α*-glucosidase (**A**) and *α*-amylase inhibitory activity (**B**) of free and bound fractions of four highland barley varieties (mean ± SD, *n* = 3).

**Figure 3 ijms-21-01175-f003:**
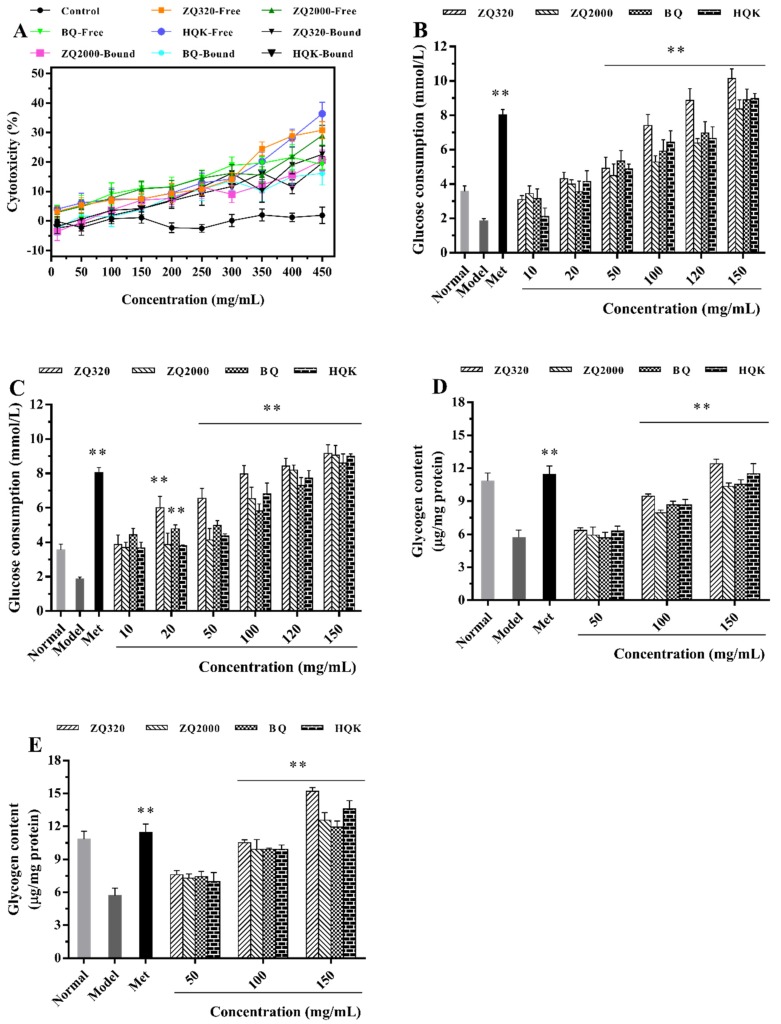
Cytotoxicity (**A**), glucose consumption of HepG2 cells in free (**B**) and bound (**C**) fractions, and glycogen accumulation of HepG2 cells in free (**D**) and bound (**E**) fractions of four highland barley varieties (mean ± SD, *n* = 3). ** Correlation is significant at the 0.01 level (2-tailed).

**Figure 4 ijms-21-01175-f004:**
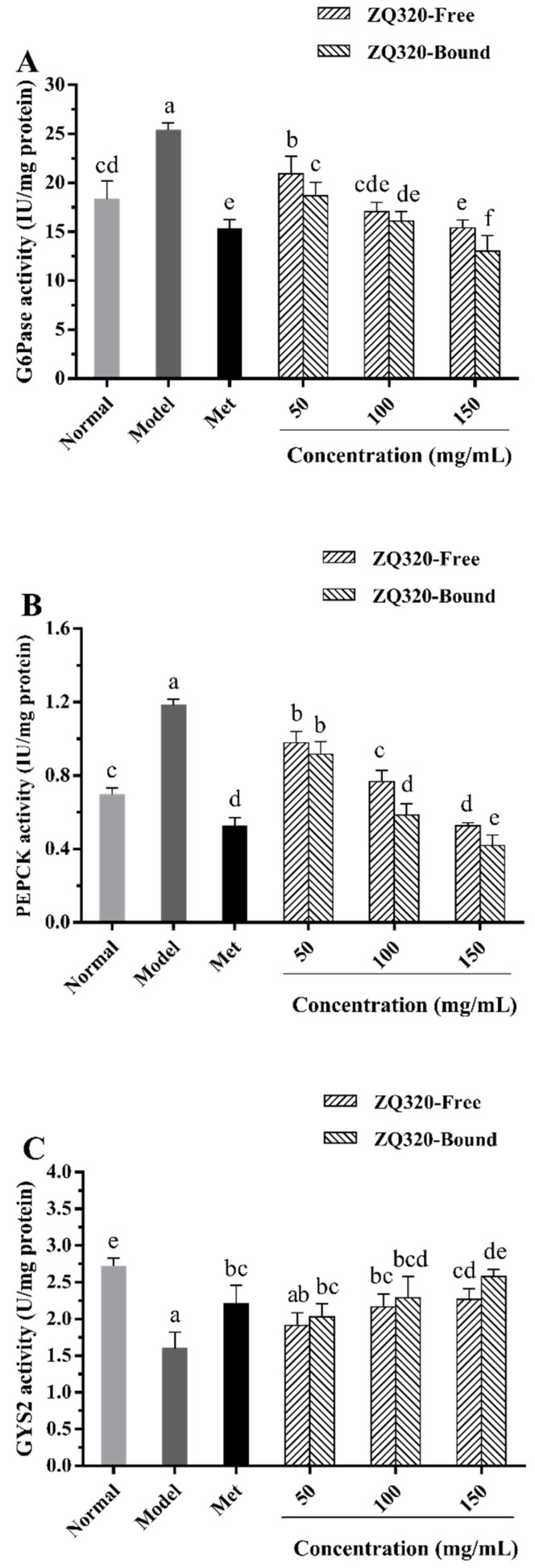
The concentration of **glucose-**6-phosphatase (G6Pase) (**A**), phosphoenolpyruvate carboxykinase (PEPCK) (**B**), and glycogen synthase 2 (GYS2) (**C**) activities of HepG2 cells in free and bound fractions of ZQ320 highland barley variety (mean ± SD, *n* = 3). Values with different letters in each bar differ significantly at *p* < 0.05.

**Figure 5 ijms-21-01175-f005:**
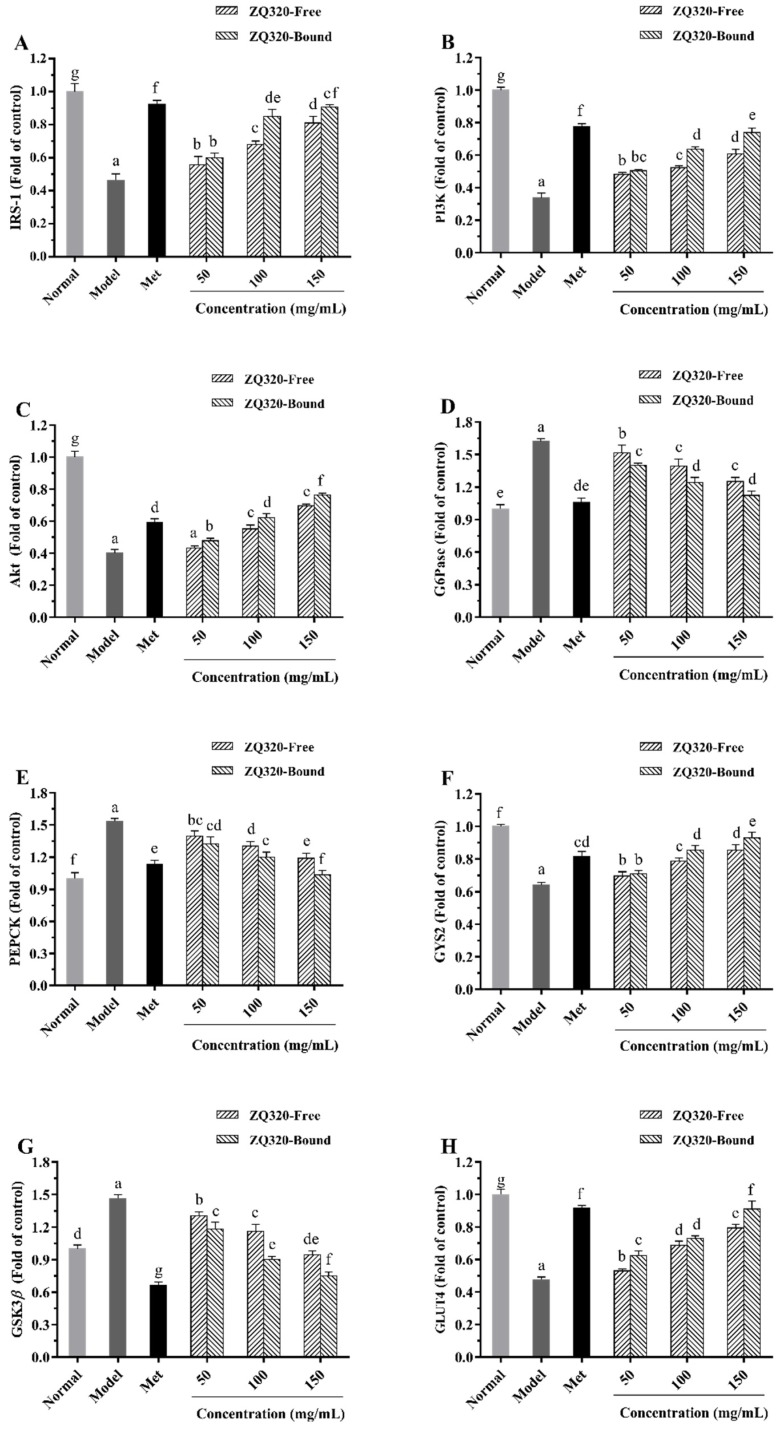
The gene expression of insulin receptor substrate-1 (IRS-1) (**A**), phosphatidylinositol 3 kinase (PI3K) (**B**), serine/threonine kinase (Akt) (**C**), G6Pase (**D**), PEPCK (**E**), GYS2 (**F**), glycogen synthase kinase 3β (GSK3β) (**G**), and glucose transporter type 4 (GLUT4) (**H**) of HepG2 cells in free and bound fractions of ZQ320 highland barley variety (mean ± SD, *n* = 3). Values with different letters in each bar differ significantly at *p* < 0.05.

**Table 1 ijms-21-01175-t001:** Moisture and total anthocyanin contents, and median cytotoxic concentration (CC_50_) of cytotoxicity in four highland barley varieties (mean ± SD, *n* = 3).

Varieties	Moisture Content (%)	Fractions	Total Anthocyanin Content (%)	CC_50_ of Cytotoxicity (mg/mL)
ZQ320 ^A^/White	12.66 ± 0.05b	Free	0.42 ± 0.08b (100) ^B^	> 600.0
		Bound	ND ^C^	538.2 ± 8.4
		Total	0.42 ± 0.08b	-
ZQ2000/White	11.95 ± 0.05d	Free	0.47 ± 0.09b (100)	> 600.0
		Bound	ND	384.5 ± 17.5
		Total	0.47 ± 0.09b	-
BQ/Blue	14.44 ± 0.03a	Free	0.86 ± 0.02b (100)	> 600.0
		Bound	ND	556.4 ± 29.8
		Total	0.86 ± 0.02b	-
HQK/Black	12.79 ± 0.10c	Free	9.24 ± 0.79a (100)	> 600.0
		Bound	ND	349.1 ± 27.8
		Total	9.24 ± 0.79a	-

Values with different lowercase letters in each column differ significantly at *p* < 0.05. ^A^ ZQ320, Zangqing 320; ZQ2000, Zangqing 2000; BQ, Bulihuang; HQK, Heiqingke. ^B^ Values in parentheses indicate percentage contribution of this fraction to the corresponding total fraction. ^C^ ND, not detected.

**Table 2 ijms-21-01175-t002:** Phenolic profiles of four highland barley varieties (mg phenolics/100 g grain, dry weight (DW); mean ± SD, *n* = 3).

Varieties	Fractions	Phenolic acids (mg/100 g DW)				Flavonoids (mg/100 g DW)		
		Gallic acid	Protocatechuic acid	Chlorogenic acid	Ferulic acid	Naringin	Catechin	Quercetin
ZQ320 ^A^	Free	0.47 ± 0.00b (100) ^B^	1.89 ± 0.01d (100)	0.97 ± 0.04d (49.36)	0.10 ± 0.00c (0.20)	0.13 ± 0.00f (1.54)	3.07 ± 0.08c (100)	ND
	Bound	ND ^C^	ND	0.99 ± 0.02d (50.64)	50.06 ± 1.46b (99.80)	8.31 ± 0.37c (98.46)	ND	2.21 ± 0.03b (100)
	Total	0.47 ± 0.00b	1.89 ± 0.01d	1.96 ± 0.02a	50.16 ± 1.45b	8.44 ± 0.37c	3.07 ± 0.08c	2.21 ± 0.03b
ZQ2000	Free	1.38 ± 0.02a (100)	1.53 ± 0.04e (100)	0.99 ± 0.04d (53.51)	0.13 ± 0.00c (0.24)	0.16 ± 0.01f (1.85)	2.47 ± 0.01d (100)	ND
	Bound	ND	ND	0.86 ± 0.02e (46.49)	54.52 ± 0.78a (99.76)	8.38 ± 0.10c (98.15)	ND	3.00 ± 0.08a (100)
	Total	1.38 ± 0.02a	1.53 ± 0.04e	1.85 ± 0.06b	54.65 ± 0.77a	8.54 ± 0.11c	2.47 ± 0.01d	3.00 ± 0.08a
BQ	Free	0.13 ± 0.01c (100)	3.21 ± 0.13b (100)	0.97 ± 0.03d (100)	ND	0.63 ± 0.04e (6.41)	3.10 ± 0.04c (65.72)	ND
	Bound	ND	ND	ND	51.18 ± 1.53b (100)	9.21 ± 0.26b (93.59)	1.62 ± 0.02g (34.28)	1.40 ± 0.03d (100)
	Total	0.13 ± 0.01c	3.21 ± 0.13b	0.97 ± 0.03d	51.18 ± 1.53b	9.84 ± 0.23a	4.72 ± 0.01a	1.40 ± 0.03d
HQK	Free	0.16 ± 0.02c (100)	2.69 ± 0.07c (64.57)	1.23 ± 0.01c (100)	ND	0.24 ± 0.04ef (3.30)	1.89 ± 0.16f (47.35)	ND
	Bound	ND	1.48 ± 0.02e (35.43)	ND	51.13 ± 2.76b (100)	7.08 ± 0.30d (96.70)	2.11 ± 0.07e (52.65)	1.81 ± 0.04c (100)
	Total	0.16 ± 0.02c	4.17 ± 0.06a	1.23 ± 0.01c	51.13 ± 2.76b	7.32 ± 0.32d	4.00 ± 0.16b	1.81 ± 0.04c

Values with different lowercase letters in each column differ significantly at *p* < 0.05. ^A^ ZQ320, Zangqing 320; ZQ2000, Zangqing 2000; BQ, Bulihuang; HQK, Heiqingke. ^B^ Values in parentheses indicate percentage contribution of this fraction to the corresponding total fraction. ^C^ ND, not detected.

**Table 3 ijms-21-01175-t003:** The median inhibition dose (IC_50_) of *α*-glucosidase and *α*-amylase inhibitions on four highland barley varieties (mean ± SD, *n* = 3).

Varieties	Fractions	IC_50_ of *α*-glucosidase Inhibition (mg/mL)	IC_50_ of *α*-amylase Inhibition (mg/mL)
ZQ320 ^A^/White	Free	176.0 ± 10.1	ND ^B^
	Bound	20.29 ± 0.53	9.48 ± 1.04
ZQ2000/White	Free	164.1 ± 22.1	ND
	Bound	27.05 ± 1.09	11.13 ± 0.34
BQ/Blue	Free	90.10 ± 5.37	ND
	Bound	43.31 ± 0.46	69.76 ± 5.02
HQK/Black	Free	80.43 ± 6.00	ND
	Bound	62.99 ± 2.82	105.4 ± 1.7
Acarbose		4.62 ± 0.68	3.36 ± 0.47
Ferulic acid		10.80 ± 0.90	7.61 ± 0.83
Naringin		17.06 ± 0.62	49.75 ± 0.47
Catechin		8.04 ± 0.50	7.04 ± 0.39

Values with different lowercase letters in each column differ significantly at *p* < 0.05. ^A^ ZQ320, Zangqing 320; ZQ2000, Zangqing 2000; BQ, Bulihuang; HQK, Heiqingke. ^B^ ND, not detected.

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
