# Peer review of "Assessment of the Phenolic Profiles, Hypoglycemic Activity, and Molecular Mechanism of Different Highland Barley (Hordeum vulgare L.) Varieties"

_ijms, 2020, doi:10.3390/ijms21041175_

Round 1

Reviewer 1 Report

General comment: The research article entitled “Assessment of Phenolic Profiles, Hypoglycemic Activity and the Molecular Mechanism of  Different Highland Barley (Hordeum vulgare L.) Varieties” is a well-organized study, with sufficient methodology and adequate description of the results. Some minor corrections are required for the improvement of the manuscript.

Abstract: The Abstract is well written and adequately presents the aim and the basic results of the study.

-Line 12. Authors could add “and the molecular mechanism of” the phrase “the effect on T2DM”

Introduction: The introduction section is well-written and adequately covers the importance of barley study on T2DM.

Materials and Methods:  The materials and methods are adequately presented. 

Results: The results of the study are analytically presented. Tables and Figures are adequate explain the findings of the study.

-Figures 2 and 3 should be represented in more clear form.

Discussion: The results of study are sufficiently discussed.

-Line 97. The paragraph could divided to 2 paragraphs.

Conclusion: The conclusion is adequate and summarizes the main text.

Bibliography/References: The references used by the authors cover adequately the relative scientific field and the aims of the study.

Author Response

Dear editor,

On behalf of the co-authors, we thank you very much for giving us an opportunity to revise our manuscript. We appreciated you and reviewer very much for the positive and constructive comments and suggestions on our manuscript. We have studied the comments carefully and made corresponding revisions by points as in the “detailed response to reviewers”.

We would like to express our great appreciation to you and reviewers for comments on our paper. Looking forward to hear from you soon.

Thank you and best regards,

Yours sincerely,

Na Deng & Bisheng Zheng

Detailed Response to Reviewer 1:

Comments and Suggestions for Authors

General comment: The research article entitled “Assessment of Phenolic Profiles, Hypoglycemic Activity and the Molecular Mechanism of Different Highland Barley (Hordeum vulgare L.) Varieties” is a well-organized study, with sufficient methodology and adequate description of the results. Some minor corrections are required for the improvement of the manuscript.

Abstract: The Abstract is well written and adequately presents the aim and the basic results of the study.

-Line 12. Authors could add “and the molecular mechanism of” the phrase “the effect on T2DM”

Response:

Thank you for your review and comments. We have added the phrase “the effect on T2DM” as suggested (Line 12). Thank you.

Introduction: The introduction section is well-written and adequately covers the importance of barley study on T2DM.

Materials and Methods: The materials and methods are adequately presented.

Results: The results of the study are analytically presented. Tables and Figures are adequate explain the findings of the study.

-Figures 2 and 3 should be represented in more clear form.

Response:

Thank you for your suggestion. Figures 2 and 3 have been represented in more clear form (1200 dpi).

Discussion: The results of study are sufficiently discussed.

-Line 97. The paragraph could be divided to 2 paragraphs.

Response:

Thank you for your careful review. The paragraph (Line 105) has been divided to 2 paragraphs as suggested. Thank you very much.

Conclusion: The conclusion is adequate and summarizes the main text.

Bibliography/References: The references used by the authors cover adequately the relative scientific field and the aims of the study.

Reviewer 2 Report

Manuscript presents a study that suggests highland barley as an effective candidate for the prevention and treatment of T2DM to improve human health due to phenolic content. It contains novel information and in general paper is well written and presented without spelling or grammar mistakes. The problem and the study of it is well discussed and conclussions are supported by results.

Some minor queries exist that are needed to be answered, please respond to the following:

p. 14, L. 396: revise "mini-Q-water" to "Milli-Q-water" Concerning HPLC analysis why authors monitored phenolics only at 280 nm? Quercetin for example has a major absorption at 380 nm and chlorogenic acid at 330nm. Please comment on this. Figure 1: Bottom error bars are not vivible, hence significant differences cannot be checked, please revise. table 1: some results for IC50 are expressed with one and some other by two decimal digits (e.g. for ZQ320 A/White 9.48 ± 1.04 and for ZQ2000/White 11.1 ± 0.3. Please check significant digits depending on the importance to have one or two decimal digits for this method. 5. table 2: as mentioned above please recheck significant digits for results expression, also the LOD and LOQ of method are needed to be presented. Did authors perform spiked samples analysis? Figure 4: please add bottom error bars, so as to check significant differences. the same for figure 5. references: some of them are more than 10 years old please update with more recent one if possible.

Author Response

Dear editor,

On behalf of the co-authors, we thank you very much for giving us an opportunity to revise our manuscript. We appreciated you and reviewer very much for the positive and constructive comments and suggestions on our manuscript. We have studied the comments carefully and made corresponding revisions by points as in the “detailed response to reviewers”.

We would like to express our great appreciation to you and reviewers for comments on our paper. Looking forward to hear from you soon.

Thank you and best regards,

Yours sincerely,

Na Deng & Bisheng Zheng

Detailed Response to Reviewer 2:

Comments and Suggestions for Authors

Manuscript presents a study that suggests highland barley as an effective candidate for the prevention and treatment of T2DM to improve human health due to phenolic content. It contains novel information and in general paper is well written and presented without spelling or grammar mistakes. The problem and the study of it is well discussed and conclusions are supported by results.

Some minor queries exist that are needed to be answered, please respond to the following:

14, L. 396: revise "mini-Q-water" to "Milli-Q-water" Concerning HPLC analysis why authors monitored phenolics only at 280 nm? Quercetin for example has a major absorption at 380 nm and chlorogenic acid at 330nm. Please comment on this. Figure 1: Bottom error bars are not vivible, hence significant differences cannot be checked, please revise. table 1: some results for IC50 are expressed with one and some other by two decimal digits (e.g. for ZQ320 A/White 9.48 ± 1.04 and for ZQ2000/White 11.1 ± 0.3. Please check significant digits depending on the importance to have one or two decimal digits for this method. 5. table 2: as mentioned above please recheck significant digits for results expression, also the LOD and LOQ of method are needed to be presented. Did authors perform spiked samples analysis? Figure 4: please add bottom error bars, so as to check significant differences. the same for figure 5. references: some of them are more than 10 years old please update with more recent one if possible.

Response:

Thank you for your careful review and valuable suggestions.

We have revised “Milli-Q water” to “Milli-Q-water” as suggested (Line 403, 406, 420).

According to our reference (Free and bound phenolic compound content and antioxidant activity of different cultivated blue highland barley varieties from the Qinghai-Tibet Plateau. Molecules 2018, 23, 879, doi:10.3390/molecules23040879.) and other papers (E.g. Relationship between phenolic compounds, anthocyanins content and antioxidant activity in colored barley germplasm. J. Agric. Food Chem., 2007, 55(12), 4802-4809, doi:10.1021/jf0701943), the phenolic profiles of barley were monitored only at 280 nm and the detected phenolics also including quercetin and chlorogenic acid. Therefore, we detected phenolics at 280 nm.

Figure 1 has been updated in a higher resolution (1200 dpi).

Based on the principle of significant digits and other published papers (E.g. Phenolic contents and cellular antioxidant activity of Chinese hawthorn “crataegus pinnatifida”. Food Chem., 2015, 186, 54-62, doi:10.1016/j.foodchem.2015.03.017; Phytochemical profiles and cellular antioxidant activity of malus doumeri (bois) chevalier on 2,2′-azobis (2-amidinopropane) dihydrochloride (ABAP)-induced oxidative stress. J. Funct. Foods, 2016, 25, 242-256, doi: 10.1016/j.jff.2016.06.004), all the authors think that the decimal digits of Table 1 and Table 2 are correct.

The LOD and LOQ of method were presented in Table S1 as suggested.

We performed the spiked sample analysis, the spiked recoveries ranged from 82.86 ± 3.21 % to 102.41 ± 1.37 % (Line 162), and these data have been shown in supplementary files (Table S1).

Figures 4 and 5 have been revised in a higher resolution (1200 dpi).

The references have been updated as suggested (E.g. Line 509, 512, 608). Thank you very much.

Reviewer 3 Report

Reviewer Evaluation:

This research paper assesses how the phenolic compounds and composition of different highland barley affect their hypoglycemic activity  and it tries to unravel the mechanism involved. This paper had quite an interesting approach with varied methods. Overall the manuscript is well written but requires minor syntax adjustments. There are also some questions to consider.

General Comments:

The introduction is well written, but there is no mention of what is currently known/unknown about the molecular pathways that these phenolic compounds affect. It would be good to clarify this in a short few sentences to focus the research paper. Line 32 – just for your own interest and not relevant to the article, there may be a type 3 diabetes (https://www.sciencedirect.com/science/article/pii/S0925443916302150). Line 201 - Can you speculate why the bound phenolic fractions had a more favourable inhibitory capacity? Generally speaking, IR and T2DM is also a pro-inflammatory disease, it would be very interesting to see if these extracts exhibited anti-inflammatory effects in the future. The title needs slight adjustment as follows: Assessment of the Phenolic Profiles, Hypoglycemic Activity and Molecular Mechanisms of Different Highland Barley (Hordeum vulgare ) Varieties

Major Points of improvement:

There are five references to ‘data not shown’. Please support the manuscript with the data either in a supplementary file or with a citation to published research. It is incorrect to make reference to undisclosed data so frequently in the discussion of the current results as it tends to do the opposite of supporting your current results and it takes away from the value of your current data. Could you please provide a clearer figure 3 and 5, thank you. You have conducted HPLC, please provide the chromatograms at least in the supplemental information or a citation to where they can be found if previously conducted. The main issue I have is that these studies are mainly in vitro and there is a sense of over extrapolation of the data for the treatment of diabetes. I suggest toning this down slightly by including a limitations section in the discussion or conclusion.

Minor Points:

Line 31 – “Clinically, DM is …”

Lines 43-44 – grammar issues

Line 45 – …classified as hulled barley with husks and hulless barley without husks.

Line 46 – replace ‘existed’ with present

Line 47 – …researchers…

Line 52 – Emerging evidence supports the notion that the antidiabetic…

Line 54 – are present in both free and bound forms.

Line 54 – Bound phenolics are able to …

Line 87 – …the bound fraction

Line 88 – you mean ‘dominant fraction’?

Line 94 – might stem…

Line 97 – belonging to …

Line 106 – which showed a similar trend to the phenolics.

Line 120 – which was in accordance with

Line 120-121 – grammar issues

Line 164 – do you mean… This may be due to variation in their genetics, environmental factors, and methodological differences.

Line 165 - …imply…

Line 206 – researchers

Line 229 – delete ‘the’

Line 241 – replace ‘prohibited’ with ‘inhibited’

Line 249 – IR is a state of hepatic, adipose, and muscle cells/tissues…

Line 251 – The liver…

Line 261 - supplementation

Line 266 …the bound fractions…

Line 268 - … were inverse to that of the…

Line 312 – replace eliminate with reduce

Line 336 – Delete concretely

Line 374 – replace executed with measured

Line 378 – Phenolics from the samples were …

Line 47 – was conducted as previously reported

Concluding Remarks:

Overall, this research is very interesting but requires minor adjustments. I enjoyed reading your paper and look forward to seeing the adjustments

Author Response

Dear editor,

On behalf of the co-authors, we thank you very much for giving us an opportunity to revise our manuscript. We appreciated you and reviewer very much for the positive and constructive comments and suggestions on our manuscript. We have studied the comments carefully and made corresponding revisions by points as in the “detailed response to reviewers”.

We would like to express our great appreciation to you and reviewers for comments on our paper. Looking forward to hear from you soon.

Thank you and best regards,

Yours sincerely,

Na Deng & Bisheng Zheng

Detailed Response to Reviewer 3:

Comments and Suggestions for Authors

Reviewer Evaluation:

This research paper assesses how the phenolic compounds and composition of different highland barley affect their hypoglycemic activity and it tries to unravel the mechanism involved. This paper had quite an interesting approach with varied methods. Overall the manuscript is well written but requires minor syntax adjustments. There are also some questions to consider.

General Comments:

The introduction is well written, but there is no mention of what is currently known/unknown about the molecular pathways that these phenolic compounds affect. It would be good to clarify this in a short few sentences to focus the research paper. Line 32 – just for your own interest and not relevant to the article, there may be a type 3 diabetes (https://www.sciencedirect.com/science/article/pii/S0925443916302150). Line 201 - Can you speculate why the bound phenolic fractions had a more favourable inhibitory capacity? Generally speaking, IR and T2DM is also a pro-inflammatory disease, it would be very interesting to see if these extracts exhibited anti-inflammatory effects in the future. The title needs slight adjustment as follows: Assessment of the Phenolic Profiles, Hypoglycemic Activity and Molecular Mechanisms of Different Highland Barley (Hordeum vulgare L.) Varieties.

Response:

Thank you for your careful review and valuable comments.

The molecular pathways of these detected phenolic compounds on T2DM have been described in Introduction section (Line 64-69).

The classification of diabetes mellitus has been updated in the text (Line 33).

The reason that bound phenolic fractions had a more favourable inhibitory capacity might be ascribed to their rich contents of ferulic acid and naringin, which previous studies suggested ferulic acid and naringin as competitive suppressors of α-glucosidase (Line 220-224).

The anti-inflammatory effects of these phenolic extracts will be explored in our future work.

The title has been revised as suggested (Line 2).

Major Points of improvement:

There are five references to ‘data not shown’. Please support the manuscript with the data either in a supplementary file or with a citation to published research. It is incorrect to make reference to undisclosed data so frequently in the discussion of the current results as it tends to do the opposite of supporting your current results and it takes away from the value of your current data. Could you please provide a clearer figure 3 and 5, thank you. You have conducted HPLC, please provide the chromatograms at least in the supplemental information or a citation to where they can be found if previously conducted. The main issue I have is that these studies are mainly in vitro and there is a sense of over extrapolation of the data for the treatment of diabetes. I suggest toning this down slightly by including a limitations section in the discussion or conclusion.

Response:

Thank you for your valuable suggestions.

The data about “data not shown” section has been summarized in Table 1, Table 3 and Table S1.

Figures 3 and 5 have been updated in a higher resolution (1200 dpi) as suggested.

The HPLC chromatograms of samples were depicted in supplementary files (Figure S1).

The limitations of in vitro studies have been discussed in the Results and Discussion section (Line 363-366, 381-382).

Minor Points:

Line 31 - “Clinically, DM is …”

Lines 43-44 - grammar issues

Line 45 - …classified as hulled barley with husks and hulless barley without husks.

Line 46 - replace ‘existed’ with present

Line 47 - …researchers…

Line 52 - Emerging evidence supports the notion that the antidiabetic…

Line 54 - are present in both free and bound forms.

Line 54 - Bound phenolics are able to …

Line 87 - …the bound fraction

Line 88 - you mean ‘dominant fraction’?

Line 94 - might stem…

Line 97 - belonging to …

Line 106 - which showed a similar trend to the phenolics.

Line 120 - which was in accordance with

Line 120-121 - grammar issues

Line 164 - do you mean… This may be due to variation in their genetics, environmental factors, and methodological differences.

Line 165 - …imply…

Line 206 - researchers

Line 229 - delete ‘the’

Line 241 - replace ‘prohibited’ with ‘inhibited’

Line 249 - IR is a state of hepatic, adipose, and muscle cells/tissues…

Line 251 - The liver…

Line 261 - supplementation

Line 266 - …the bound fractions…

Line 268 - … were inverse to that of the…

Line 312 - replace eliminate with reduce

Line 336 - Delete concretely

Line 374 - replace executed with measured

Line 378 - Phenolics from the samples were …

Line 437 - was conducted as previously reported

Response:

Thank you for your careful review. All the phrases have been revised in the text as suggested. Thank you very much.

Concluding Remarks:

Overall, this research is very interesting but requires minor adjustments. I enjoyed reading your paper and look forward to seeing the adjustments.